# Comparison of Iron Oxide Nanoparticles in Photothermia and Magnetic Hyperthermia: Effects of Clustering and Silica Encapsulation on Nanoparticles' Heating Yield

**Sebastjan Nemec** [1,2,†], **Slavko Kralj** [1,2,*,†] , **Claire Wilhelm** [3], **Ali Abou-Hassan** [4] ,
**Marie-Pierre Rols** [5] and **Jelena Kolosnjaj-Tabi** [5,*]

1   Department for Materials Synthesis, Jožef Stefan Institute, Jamova cesta 39, 1000 Ljubljana, Slovenia; sebastjan.nemec@ijs.si
2   Faculty of Pharmacy, University of Ljubljana, Aškerčeva cesta 7, 1000 Ljubljana, Slovenia
3   Laboratoire Matière et Systèmes Complexes (MSC), UMR 7057, CNRS and Université Paris Diderot, Bâtiment Condorcet, 10 rue Alice Domon et Léonie Duquet, 75205 Paris, France; claire.wilhelm@univ-paris-diderot.fr
4   Sorbonne Université, CNRS UMR 8234, PHysico-chimie des Electrolytes et Nanosystèmes InterfaciauX (PHENIX), F-75005 Paris, France; ali.abou_hassan@sorbonne-universite.fr
5   Institute of Pharmacology and Structural Biology, 205 Route de Narbonne, 31400 Toulouse, France; Marie-Pierre.Rols@ipbs.fr
*   Correspondence: slavko.kralj@ijs.si (S.K.); jelena.kolosnjaj-tabi@ipbs.fr (J.K.-T.)
†   These authors contributed equally to this work.

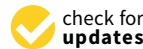

**Featured Application: Superparamagnetic iron oxide nanoparticles (SPIONs) have a recognized potential for magnetic hyperthermia, and they are also being increasingly proposed as agents for photothermal treatment (photothermia), a biomedical modality where nanoparticles are excited by light to generate local hyperthermia. While it is known that endosomal internalization of SPIONs negatively affects magnetic hyperthermia, photothermia is not decreased. In an attempt to mimic nanoparticles clustering in endosomes, we herein investigate the effects of silica encapsulation and SPION clustering on both magnetic hyperthermia and photothermia.**

**Abstract:** Photothermal therapy is gathering momentum. In order to assess the effects of the encapsulation of individual or clustered superparamagnetic iron oxide nanoparticles (SPIONs) on nanoparticle light-to-heat conversion, we designed and tested individual and clustered SPIONs encapsulated within a silica shell. Our study compared both photothermia and magnetic hyperthermia, and it involved individual SPIONs as well as silica-encapsulated individual and clustered SPIONs. While, as expected, SPION clustering reduced heat generation in magnetic hyperthermia, the silica shell improved SPION heating in photothermia.

**Keywords:** superparamagnetic iron oxide nanoparticles; silica-coated magnetic nanoparticles clusters; hyperthermia; photothermal treatment; encapsulation

---

## 1. Introduction

Matter can reflect, transmit, or absorb light. The reflected or transmitted light is scattered in space, while the optical energy of absorbed light can be converted to heat. The heating, which occurs when light interacts with nanomaterials, can have different physical mechanisms, which depend on the nature of the nanomaterial [1]. In metals, such as silver, copper, or gold, heat occurs when light

interacts with conduction electrons on the nanoparticles' surface. In semiconducting materials, such as iron oxides, optical radiation energy allows a temporary transition of electrons from the valence band to the conduction band, which results in heat generation when the electrons relax back to the valence band [1].

Heat (or hyperthermia) is an attractive physical approach that can be used in cancer treatment [2,3]. When the temperature in tissues increases, typically between 40 and 45 °C, the blood flow and the oxidation of tissues increase [4], collagen fibers slacken [5], and tumors become more sensitive to chemotherapeutics [6,7] or radiation [8,9].

Currently, there are different clinical approaches which employ probes and needles to generate heat by microwaves, radiofrequencies, or ultrasound [10,11]. While efficient, these approaches are invasive and may be detrimental to bystander tissue. On the other hand, promising preclinical approaches were reported in which nanoparticles were used to heat tumor tissues [5,12–14].

Among the different approaches, the nanoparticle light-to-heat conversion principle represents an excellent therapeutic alternative to current approaches and is particularly attractive for several reasons. First, nanoparticle suspensions can simply be injected into desired tissues, which does not require surgical intervention. Secondly, near-infrared light (650 to 1350 nm) may penetrate the tissue, meaning that light can be used to trigger heat generation in injected nanoparticles. Thirdly, the bystander tissue, between the light source and the target, can remain undamaged.

Photothermal treatment mediated by plasmonic (gold) nanoparticles already reached clinical trials (NCT01679470, NCT01270139), and other materials [15,16] are emerging for photothermia in preclinical research. Among the different materials, iron oxides appear to be excellent candidates not only because they effectively generate heat under different physical modalities [17], but also because they can be used as diagnostic tools and are magneto-responsive [18,19] and could thus be magnetically guided to the site of interest [20]. Iron oxides are also safe [21] and biocompatible [22], and they were already used for magnetic hyperthermia in clinical settings [23,24]. While several research groups reported the efficacy of photo-induced heating of iron oxides [19,25–27], little is known about the effect of iron oxide's silica encapsulation and superparamagnetic iron oxide nanoparticles (SPIONs) clustering on the photothermal yield.

In magnetic hyperthermia, intracellular confinement is now recognized as an important issue because nanoparticle aggregation within endosomes inhibits both physical mechanisms that account for heat generation: nanoparticle Brownian relaxation—the rotation of entire magnetic nanoparticles within their surroundings—and Néel relaxation, or the rotation of the magnetic moment within the magnetic cores [17]. However, when different types of iron oxide nanoparticles are internalized in endosomes, their photothermal yield can slightly increase, even in comparison to nanoparticle suspensions in water [17]. With the aim of evaluating the effect of SPIONs' encapsulation and clustering on the heating outcome, we present a study in which we compare the heating efficacy of non-encapsulated, individually encapsulated, or clustered and encapsulated SPIONs. The heating is compared in photothermal regime and in magnetic hyperthermia.

## 2. Materials and Methods

### 2.1. Materials

All chemicals used for the synthesis were of reagent grade quality and were obtained from commercial sources. The magnetic nanoparticle clusters were provided by Nanos SCI (Ljubljana, Slovenia) and are commercially available as iNANOvative™. Iron (III) sulphate hydrate, iron(II) sulphate heptahydrate (ACS, 99%), citric acid (99%), and $NH_4OH$ (25%) were supplied by Alfa Aesar (Lancashire, UK). Polyacrylic acid (PAA, 30 wt% solution, MW 30 kDa) was purchased from PolySciences GmbH, Hirschberg an der Bergstrasse, Germany. Acetone (AppliChem GmbH, Darmstadt, Germany) and ethanol absolute (Carlo Erba, reagent, USP, Milan, Italy) were used without further

processing. Tetraethoxysilane (TEOS; 98%) and polyvinyl pyrrolidone (PVP, 40 kDa) were obtained from Sigma Aldrich (St. Louis, MO, USA).

## 2.2. Superparamagnetic Iron Oxide Nanoparticles (SPIONs) Synthesis

The superparamagnetic iron oxide nanoparticles (SPIONs), in the form of maghemite ($\gamma$-$Fe_2O_3$), were prepared by co-precipitation of $Fe^{2+}$ and $Fe^{3+}$ ions from an aqueous solution, as reported elsewhere [28–31]. Briefly, ferrous sulfate ($FeSO_4$) and ferric sulfate ($Fe_2(SO_4)_3$) were dissolved in distilled water in order to obtain final concentrations of 0.027 mol/L of $Fe^{2+}$ and 0.014 mol/L of $Fe^{3+}$. Then, precipitation was triggered with aqueous ammonia (~25%) in two steps. First, the pH was adjusted to 3 and kept constant at that value for 30 min. After that time period, the pH was set to 11.6. After additional 30 min, the formed nanoparticles were collected with a magnet and washed five times with an aqueous ammonia solution at pH 10.5, then finally dispersed in 120 mL of water. The washed nanoparticles were further functionalized with citric acid. A volume of 5 mL of a 0.5 g/mL citric acid aqueous solution was added to the nanoparticle suspension in 120 mL of water, and the pH was adjusted to 5.2 with aqueous ammonia. The reaction mixture was then magnetically stirred at 450 rpm in an oil bath at 80 °C for 90 min. After that, the pH was set to 10.2 with aqueous ammonia. Finally, the obtained suspension was centrifuged at 8000 $g$ for 5 min to remove and discard any sediment of aggregated nanoparticles (approximately 3–5% of all nanoparticles) while the supernatant, representing the colloidal suspension, was used for further procedures.

## 2.3. SPIONs Silica Encapsulation (SPIONs-SIL)

The citric acid-stabilized nanoparticles were encapsulated with a silica shell with a thickness of approximately 3–5 nm. The silica was deposited after hydrolysis of TEOS and the subsequent condensation of silica precursors on the surface of the nanoparticles, as described previously [28]. In summary, the pH of the aqueous suspension of the citric acid-stabilized nanoparticles (120 mL, 2 wt%) was adjusted to 11 with aqueous ammonia. Then, 2.5 mL of TEOS dissolved in 25 mL of ethanol was added. The reaction mixture was magnetically stirred at 450 rpm at room temperature overnight. The next day the silica-encapsulated nanoparticles were washed three times with ethanol and finally dispersed in water.

## 2.4. Silica-Encapsulated SPION Clusters (MNCs)

The silica-encapsulated SPION clusters were provided by Nanos Scientificae Ltd. (Nanos SCI, Ljubljana, Slovenia). These nanoparticle clusters were synthesized in a microemulsion where a large number of maghemite nanoparticles were self-assembled into spherical clusters, followed by encapsulation of the nanoparticle clusters with a layer of silica [32,33]. First, small maghemite nanoparticles were synthesized using precipitation from an aqueous solution, as described in Section 2.2. Then, the nanoparticles were self-assembled by using polyacrylic acid and polyvinylpyrrolidone. This approach was exclusively developed and is suitably protected by Nanos SCI. The nanoparticle cluster size was finally unified by the use of high-gradient magnetic separation, as described in our previous paper [32,34].

## 2.5. Materials Characterization

The nanoparticle structure was assessed by transmission electron microscopy (TEM). The drop of nanoparticle suspension was deposited on a copper grid coated with perforated carbon foil. The suspension deposited on the grid was dried prior to TEM analyses. The analyses were performed with a transmission electron microscope (Jeol JEM 2100, Jeol, Akishima, Japan) operating at 200 kV. The magnetic properties of the nanoparticles were measured at room temperature by vibrational sample magnetometry (VSM) (Lake Shore 7307 VSM, Lake Shore Cryotronics, Westervile, OH, USA). The SPION crystalline structure was analyzed by X-ray diffraction (XRD, Bruker D8 Advance, Bruker, Billerica, MA, USA, Cu K$\alpha$ radiation). The zeta potential measurements, as a function of the suspensions'

pH (at a volume of 15 mL), were carried out at a final nanoparticle concentration of 0.1 mg/mL in an aqueous solution containing KCl (final concentration of 10 mM). Zeta potential measurements were performed on Zeta PALS, Brookhaven Instruments Corporation, Holtsville, NY, USA.

The iron concentrations in the samples were determined by elemental analysis using inductively coupled plasma atomic emission spectrometry (ICP-AES) (iCAP6200 duo, Thermo Fisher Scientific, Waltham, MA, USA). Samples were dispersed in an $HNO_3$ and HCl solution (5 mL), the acid solution was evaporated, and 5 mL of a 1% HCl solution was added for the analysis. The final range of concentration was adjusted to match 10–100 ppb in iron.

### 2.6. Photothermia and Magnetic Hyperthermia Experiments

Thermal measurements were performed in 0.5 mL Eppendorf tubes, containing 50 μL of nanomaterials dispersed in water. Concentrations ranged from [Fe] = 1 mM to [Fe] = 150 mM. Samples were exposed to an 808 nm laser (Laser Components S.A.S, Meudon, France) positioned 4.5 cm above the tube, with the power density fixed at 0.3 W/cm$^2$ (corresponding to 1.82 A).

For magnetic hyperthermia modality, the same sample preparation was used. The magnetic field was generated by the nB Nanoscale Biomagnetics device (Nanoscale Biomagnetics, Zaragoza, Spain), at a frequency of 471 kHZ with an 18 mT magnetic field intensity.

For both instances, temperature elevation was recorded with an infrared thermal imaging camera (FLIR SC7000, FLIR Systems, Wilsonville, OR, USA) in real time and processed with ALTAIR software (Altair Engineering, Troy, MI, USA). Heating was quantified both with the plateau temperature reached after 5 min of laser treatment and the specific absorption rate (SAR), meaning the power dissipated per unit mass of iron (W·g$^{-1}$), according to the following equation:

$$SAR = \frac{C \times m_s}{m_{Fe}} \times \frac{dT}{dt}$$

(1)

where $C$ is the specific heat capacity of the sample ($C_{water}$ = 4.185 J/g/K), $m_{Fe}$ is the total mass of iron in the sample (g), $m_s$ is the total mass of the sample (g), and $dT/dt$ is the temperature increase at the initial linear slope (30 s).

### 2.7. Statistical Analysis

Measurements were performed in triplicate. The results are expressed as mean ± SD.

## 3. Results

### 3.1. Nanoparticles Characterization

The citrate-stabilized maghemite nanoparticles (SPIONs) (Figure 1A,B), 11.8 ± 2.4 nm in size (as determined by transmission electron microscopy (TEM), with N = 100), were encapsulated with a 3–5 nm thick (Figure 1C,D), homogeneous silica shell (SPIONs-SIL), as reported in our previous papers [28,31]. The size of the nanoparticle clusters encapsulated with approximately 6 nm thick silica shell was 95 ± 23 nm (Figure 1E,F), which was determined from the TEM micrographs (>100 clusters counted). The synthesis of the nanoparticle clusters is based on the self-assembly of N = 70 ± 14 superparamagnetic iron oxide (maghemite) nanoparticles. The core–shell nature of the SPION clusters, with the closely packed maghemite nanoparticles within the cluster cores and the amorphous silica shell, can be clearly distinguished in the TEM micrographs (Figure 1E,F).

The XRD of the precipitated nanoparticles showed a single spinel phase (Figure 2A), whereas the Mössbauer spectroscopy confirmed that the SPIONs were composed of maghemite, as reported in our previous work [32]. Both the silica-encapsulated nanoparticles and the silica-encapsulated nanoparticle clusters showed good colloidal stability (Figure 2B), while high magnetic responsiveness was observed only for the silica-coated nanoparticles clusters. Thus, when the suspension was placed near a magnet, the silica-encapsulated nanoparticle clusters rapidly migrated toward the magnet

(Figure 2B). Subsequently, as soon as the external magnet was removed, the particles could be dispersed spontaneously by a gentle shake. All nanoparticles exhibited superparamagnetic properties (Figure 2C). The saturation magnetizations ($Ms$) of the as-synthesized nanocrystals, silica-encapsulated nanoparticles, and silica-encapsulated nanoparticle clusters were 64.3, 50.3, and 53.8 emu/g, respectively (Figure 2C). While the three samples contained the same type of nanocrystals, the saturation magnetization differences among the samples originated from the content of the mass of silica per gram of nanocomposite. While as-synthesized nanocrystals have no silica shell, the saturation magnetization of the SPION-SILs, compared with that of the MNCs, decreased because SPION-SILs have a larger surface area per gram of nanocomposite and, therefore, they contain a larger portion of non-magnetic (diamagnetic) silica per gram compared with MNCs. The electrophoretic mobility of the silica-encapsulated nanoparticles and silica-encapsulated nanoparticle clusters were measured as the function of the operational pH. The silica surface showed a relatively acidic character because its structure comprised negatively charged hydroxyl groups at pH values above the isoelectric point (IEP) at pH 2.6. The zeta potential curves of the silica-encapsulated nanoparticles and silica-encapsulated nanoparticle clusters reached high negative values of zeta potential at a physiological pH of 7.4 at −29 mV and −25 mV, respectively (Figure 2D). The high absolute values of the zeta potential provided strong electrostatic repulsive forces between particles, resulting in good colloidal stability of the suspension in neutral and alkaline conditions [29].

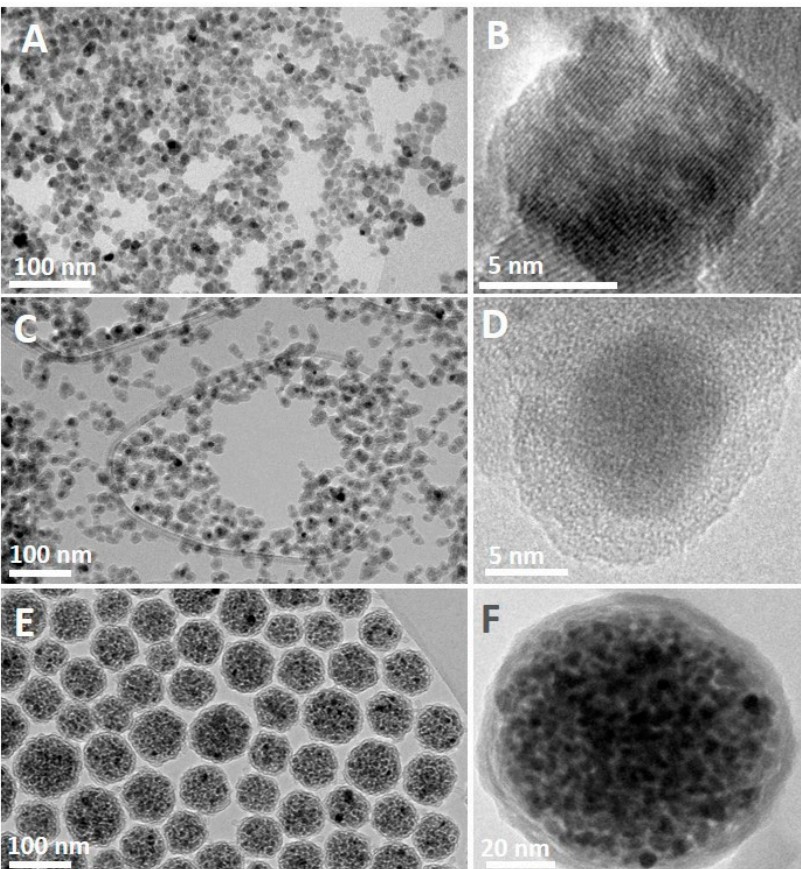

**Figure 1.** (**A**,**B**) Transmission electron micrographs of studied nanoparticles: superparamagnetic iron oxide nanoparticles (SPIONs), (**C**,**D**) silica-encapsulated individual nanoparticles (SPIONs-SIL), and (**E**,**F**) silica-encapsulated nanoparticle clusters (MNC).

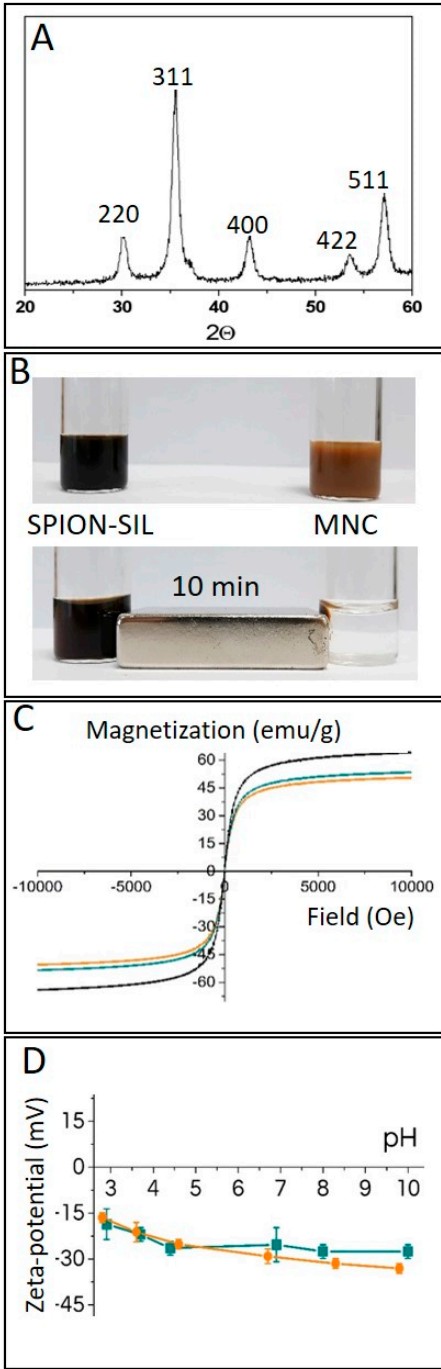

**Figure 2.** (**A**) The XRD pattern of iron oxide nanoparticles. (**B**) Photographs representing efficient magnetic separation of silica-encapsulated nanoparticle clusters (MNC), whereas silica-encapsulated individual nanoparticles (SPIONs-SIL) are not magnetically responsive. (**C**) The room temperature measurements of the specific magnetization as a function of the magnetic field for the SPIONs (black), silica-encapsulated nanoparticles (SPIONs-SIL) (orange), and MNCs (cyan). (**D**) The change of the ζ potentials of the (SPIONs-SIL) (orange) and MNCs (cyan).

### 3.2. Temperature and SAR Measurements After Photothermia and Magnetic Hyperthermia Modality

The photothermal efficiency of the three types of nanomaterials was measured in aqueous suspensions, with iron concentrations ranging from [Fe] = 1 mM to [Fe] = 150 mM. All samples were exposed to an 808 nm laser at a power density of 0.3 W/cm$^2$ for 5 min (Figure 3A–C). Typical temperature elevation curves, recorded with an IR camera, are shown in Figure 3A for all nanomaterials

at an iron concentration of [Fe] = 30 mM, and corresponded to 45.68 °C for SPIONs-SIL, 43.49 °C for SPIONs, and 39.51 °C for MNCs, respectively. This measurement allowed for retrieval of the increase of temperature ΔT reached after 5 min, which corresponds to the maximum plateau temperature. The average ΔT is shown in Figure 3B for all nanomaterial iron concentrations, and temperature increases were comparable between the SPIONs-SIL and the SPIONs (attaining a max ΔT of 20.5 °C and 20.3 °C, respectively), while the MNCs attained a max ΔT of 17.8 °C. The maximum temperature that can be reached is attained at high concentrations (in the 20–100 mM range), and further concentration increasing does not translate to a higher heating due to light absorption. This saturation of the heating with higher concentrations resulted in a decrease of the SAR, measured at increasing concentrations (Figure 3C), and converged for all nanoparticles at concentrations above 70 mM of iron.

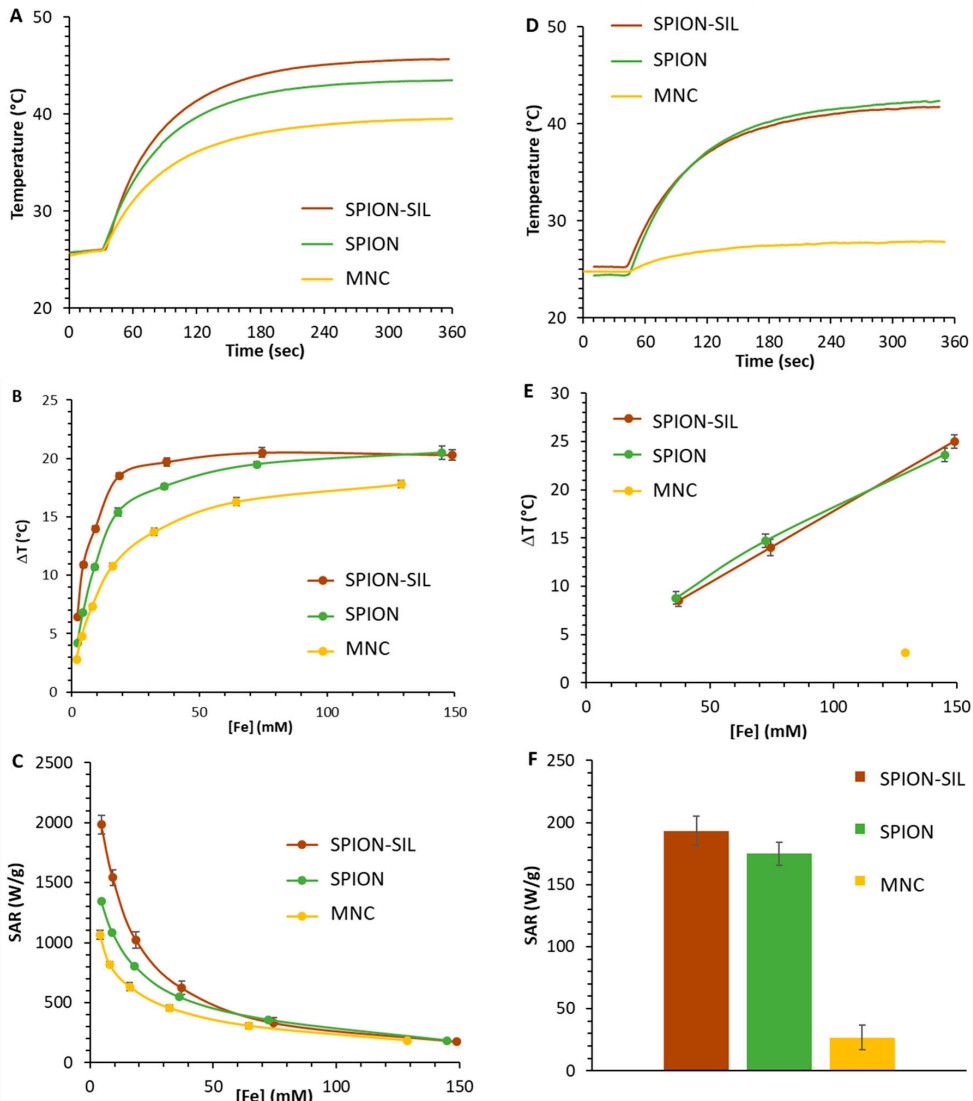

**Figure 3.** Temperature elevation measurements after (**A**–**C**) photothermia (wavelength of 808 nm, power density of 0.3 W/cm$^2$) and (**D**–**F**) magnetic hyperthermia (frequency 470 kHz, field 18 mT). (**A**) Representative temperature increase as a function of time for the suspension at iron concentration [Fe] = 30 mM; (**B**) average temperature increase as function of the iron concentration reached after 5 min of laser exposure; (**C**) average specific absorption rate (SAR) as a function of the iron concentration in different nanomaterials; (**D**) representative temperature increase as a function of time for the suspension at iron concentration [Fe] = 130 mM; (**E**) average temperature increase as a function of the concentration under magnetic hyperthermia; and (**F**) average SAR as a function of the iron concentration in different nanomaterials under magnetic hyperthermia.

Similar to photothermia, a series of magnetic hyperthermia measurements were performed when the suspensions of nanomaterials were exposed to an alternating magnetic field (frequency 470 kHz, field 18 mT). Here, temperature elevation curves were recorded at a higher concentration of iron, specifically [Fe] = 130 mM (Figure 3D), and attained temperatures of 42.38 °C for SPIONs, 41.71 °C for SPIONs-SIL, and 27.91 °C for MNCs. Figure 3E shows the average ΔT as a function of the iron concentration. These measurements were performed for three decreasing concentrations for SPIONs and SPIONs-SIL (attaining a max ΔT of 25 °C for SPIONs-SIL and 23.6 °C for SPIONs), and results could not be obtained for MNCs because, for the MNCs, the temperature elevation was not significant enough to be detected at concentrations below 100 mM. For the magnetic hyperthermia modality, the heat generation is linearly proportional to the iron concentration. As a result, the SAR is independent of the iron concentration. The average values of the SAR for different nanomaterials are provided in Figure 3F, and they are 193 W/g for SPIONs-SIL, 175 W/g for SPIONs, and 27 W/g for MNCs.

## 4. Discussion

The local surrounding (e.g., environment) of nanoparticles and nanoparticles hierarchical organization may have an impact on different therapeutic [17] and diagnostic modalities [35]. The mechanical properties of the surrounding environment [36] or the clustering SPIONs may thus affect the nanoparticles' heating yields [17]. In magnetic hyperthermia, external mechanical constraints [36], as well as nanoparticle confinement or aggregation, decreases the thermal yield because mechanically constrained or aggregated nanoparticles have less freedom to rotate (which inhibits Brownian relaxation) or because the inter-nanoparticle distance decreases, increasing nanoparticle interactions (which hampers Néel relaxation).

Upon cellular internalization, nanoparticles become confined in endosomes, and one is given as an example in Figure 4. The net result of nanoparticle confinement/aggregation is diminished heat generation in magnetic hyperthermia. Conversely, studies performed in living cells showed that, in photothermia modality, the heating does not decline upon nanoparticle internalization in cells. Nanoparticle enclosures within endosomes do not decrease the nanoparticles' capacity to heat. Importantly, intracellular aggregation showed a small increase of the nanoparticles' photothermal heating [17,27]. Based on these findings, we herein investigated whether well-defined SPION clustering, and encapsulation by a silica shell (Figure 1), might affect the photothermal yield. In this context, we tested the heating potential of individual SPIONs, individual silica-encapsulated SPIONs (SPIONs-SIL), and silica-encapsulated SPION clusters (MNCs). These MNC assemblies highly resembled endosomes (Figure 4), which are cellular compartments in which nanoparticles are circumscribed upon nanoparticle endocytosis by cells.

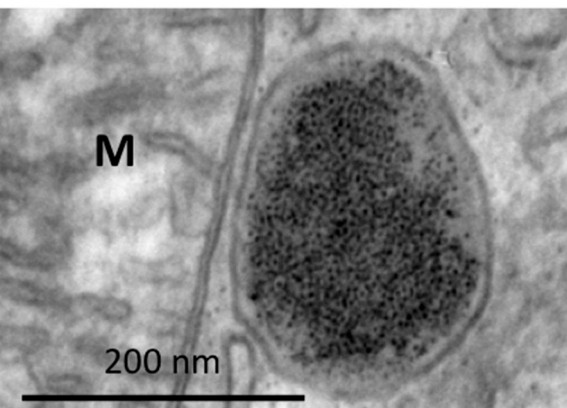

**Figure 4.** Representative transmission electron micrograph exemplifying an endosome within the cytoplasm of a cell. M denotes a mitochondrion in the vicinity of the endosome, which is laden with magnetic nanoparticles. Structurally, the nanoparticles within endosomes resemble MNCs.

In the present study, we compared two heating modalities: photothermia and magnetic hyperthermia. We aimed to provide a better understanding of the outcome of nanoparticle heating behaviors in relation to their configurations (non-encapsulated vs. encapsulated and individual vs. clustered). Indeed, all compared nanoparticle types consisted of identical maghemite core nanocrystals [32].

In photothermia, at the iron concentration of 30 mM, SPIONs-SIL heated to the highest extent (45.7 °C), followed by SPIONs (43.5 °C). Here, we first emphasize that the laser power was clinically relevant (0.3 W/cm$^2$), and second, the iron concentration was much lower than the one that is used for magnetic hyperthermia (Figure 3A). The comparison of SPIONs and SPIONs-SIL indicates that the encapsulation with silica leads to a better heating yield in this heating modality. As the nanoparticles compared in this study consist of maghemite, a potentially different crystal phase (e.g., magnetite) cannot explain the difference in nanoparticle heating. The most obvious cause for heating enhancement is thus the silica coating.

The presence of nanoparticles may increase the thermal conductivity of fluids [37], and studies show that silica nanoparticles can increase the critical heat flux [38]. Moreover, studies showed that, in water suspensions, silica-coated gold nanoparticles dissipated heat much faster than bare (citrate-stabilized) gold nanoparticles [39]. In line with these findings observed for gold nanoparticles [39], our silica-coated SPIONs also dissipated heat to a greater extent than non-encapsulated SPIONs.

In contrast, the clustering of SPIONs into well-defined, endosome-like structures (Figure 4) did not enhance photothermal heating. As MNCs are larger than SPIONs-SIL, they might dissipate heat at a slower rate [40]. In addition, the light scattering within the clusters might have resulted in a decrease in absorption efficiency of the nanoparticles.

In magnetic hyperthermia, the heating efficiencies of the SPIONs and the SPIONs-SIL were apparently similar under our experiment conditions. This can be explained in two ways. One explanation is that the silica coating of individual SPIONs does not have an impact on the mechanisms of heat generation in magnetic hyperthermia. This could be true if the heat generation is mostly due to Néel relaxation. If the heating would mainly be a result of Brownian relaxation, the increase of the nanoparticle volume due to the silica shell would decrease nanoparticle motion and thus reduce Brownian relaxation. Alternatively, our second hypothesis is that Brownian relaxation might have also been slightly hampered, thus resulting in a heating decrease. Nevertheless, this decrease was compensated by the faster heat dissipation of the silica shell, resulting in a comparable heating outcome.

In contrast, the heating of MNCs was significantly less efficient in magnetic hyperthermia, where nanoparticle clusters' heating was significantly inhibited, similar to what occurs in endosomes upon nanoparticle internalization by cells [17] and similar to what was reported in other studies comparing the heating of clustered SPIONs [41]. Within the clusters, as among the confined nanoparticles in the endosomes, the interactions between adjacent nanoparticles impair Néel relaxation, resulting in a decreased heating efficiency [17,41].

Taken together, the results of this study indicate that, with tested nanoparticle architectures, photothermia is more efficient than magnetic hyperthermia. These experimental findings were reached at laser and magnetic field settings that lay within the spectrum of clinical applicability in terms of laser power density and magnetic field frequency, respectively.

**Author Contributions:** Conceptualization, S.K. and J.K.-T.; methodology, S.K., C.W. and J.K.-T.; validation, S.K., C.W., M.-P.R. and J.K.-T.; formal analysis, S.N. and C.W.; investigation, S.N., S.K., C.W., A.A.-H.; resources, S.K. and M.-P.R.; data curation, S.K. and J.K.-T.; writing—original draft preparation, S.K., C.W. and J.K.-T.; writing—review and editing, S.N., S.K., C.W., A.A.-H., M.-P.R. and J.K.-T.; visualization, S.K. and J.K.-T.; supervision, S.K. and J.K.-T.; funding acquisition, S.K. and M.-P.R. All authors have read and agreed to the published version of the manuscript.

**Funding:** This research was funded by Slovenian Research Agency (ARRS) for Young Researcher Scheme (grant number S.N.; 1000-18-0106), Research Project (grant number J1-7302), Research Core Funding (grant number P2-0089); and by the Plan Cancer project NUMEP (ref. PC201615).

**Acknowledgments:** The authors acknowledge the CENN Nanocenter (Ljubljana, Slovenia) for the use of the electron microscope (TEM 2100). MPR and JKT thank Muriel Golzio for scientific discussions.

**Conflicts of Interest:** The authors declare no conflict of interest.

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
