# Peer review of "Comparison of Iron Oxide Nanoparticles in Photothermia and Magnetic Hyperthermia: Effects of Clustering and Silica Encapsulation on Nanoparticles’ Heating Yield"

_applsci, doi:10.3390/app10207322_

Round 1

Reviewer 1 Report

I have read this work several times and feel that it is publishable in the journal barred a few minor revisions. Largely, the only suggestion I have is that the English should be improved.

Introduction - The introduction can be improved. I suggest the authors cite the recent literature on hyperthermia and nanoparticles (i.e. doi:10.3390/nano10101919 and others).

Figure 2 - This figure can be improved. I suggest to reshape the dimensions and the arrangement in 4 rows and only one column. The font sizes need to be enlarged.

Figure 3, panel f - I suggest the authors to change the fill of the histogram in order to make the error bar more visible.

Lines 272-276 - This part of the discussion can be expanded with a more detailed comparison with the literature already cited.

Different minor typo corrections that should be performed.

Author Response

We thank the reviewer for his or her time and constructive suggestions, please refer to our point-to-point responses.

  • Introduction - The introduction can be improved. I suggest the authors cite the recent literature on hyperthermia and nanoparticles (i.e. doi:10.3390/nano10101919 and others).

We have added the suggested research and some other studies in the introduction and discussion section.

  • Figure 2 - This figure can be improved. I suggest to reshape the dimensions and the arrangement in 4 rows and only one column. The font sizes need to be enlarged.

We thank the reviewer for this suggestion. We have reorganized the figure accordingly. 

  • Figure 3, panel f - I suggest the authors to change the fill of the histogram in order to make the error bar more visible.

We thank the reviewer for the comment. We have corrected the figure. Please note that the graphs A and D represent a typical curve of temperature elevation (so no SD values are included). The SDs obtained from triplicate measurements are included in the error bars of all the other graphs, which indicate the average temperature elevation (dT) and the average SAR.

  • Lines 272-276 - This part of the discussion can be expanded with a more detailed comparison with the literature already cited.

We have expanded the discussion part with additional comparisons.

  • Different minor typo corrections that should be performed.

Thank you, we have corrected the typos.

Reviewer 2 Report

1. Comparison of magnetic heating effect in single nanoparticles and clusters, and in water suspensions and when nanoparticles are confined by tissue-mimicking structures, have been previously presented by other authors. It would be useful to mention them (presumably in the introduction or conclusion part) so that the readers know about other similar studies.

doi: 10.3390/nano9050803
doi: 10.1063/1.4939514

2. Line 139, small typo - please correct 'dispesed' to 'dispersed.

3. Line 155, you write that your results are expressed as mean and +/- SD. Could you add to the manuscript values of SD for temperature elevation measurements? The only SD value can be found on graph 3c and 3f.

4. Line 179, can you explain the difference in obtained magnetic saturation, since you used for your experiments the same type of magnetic material and magnetization is a characteristic of magnetic material.

5. Line 181, you presented saturation magnetization results in the unit [Am2kg-1] and on figure 2C in the unit [emu/g]. Can you unify the units to not to confuse less familiar with the topic readers?

6. Line 193, (b) should be in bold.

7. Line 307, please note (and correct) that in the reference section every reference has double numeration (1. 1., 2. 2.).

Author Response

We thank the reviewer for his or her time and constructive suggestions, please refer to our point-to-point responses.

  1. Comparison of magnetic heating effect in single nanoparticles and clusters, and in water suspensions and when nanoparticles are confined by tissue-mimicking structures, have been previously presented by other authors. It would be useful to mention them (presumably in the introduction or conclusion part) so that the readers know about other similar studies.

doi: 10.3390/nano9050803
doi: 10.1063/1.4939514

We thank the reviewer for pointing out these relevant studies, which we mentioned in the text in the beginning of the discussion section (for the study made by Kaczmarek et al.) and at the end of the discussion Fu et al..

  1. Line 139, small typo - please correct 'dispesed' to 'dispersed.

Thank you for noticing. We corrected the word.

  1. Line 155, you write that your results are expressed as mean and +/- SD. Could you add to the manuscript values of SD for temperature elevation measurements? The only SD value can be found on graph 3c and 3f.

We thank the reviewer for the comment. We have corrected the figure. Please note that the graphs A and D represent a typical curve of temperature elevation (so no SD values are included). The SDs obtained from triplicate measurements are included in the error bars of all the other graphs, which indicate the average temperature elevation (dT) and the average SAR.

  1. Line 179, can you explain the difference in obtained magnetic saturation, since you used for your experiments the same type of magnetic material and magnetization is a characteristic of magnetic material.

Thank you for the comment. We included explanations related to saturation magnetization differences among the samples/

  1. Line 181, you presented saturation magnetization results in the unit [Am2kg-1] and on figure 2C in the unit [emu/g]. Can you unify the units to not to confuse less familiar with the topic readers?

Thank you for pointing out this inconsistency, we have corrected the text accordingly.

  1. Line 193, (b) should be in bold.

Thank you, we corrected it.

  1. Line 307, please note (and correct) that in the reference section every reference has double numeration (1. 1., 2. 2.).

Thank you, we have corrected the numeration in References section.